# Comparative Metagenomics and Metabolomes Reveals Abnormal Metabolism Activity Is Associated with Gut Microbiota in Alzheimer’s Disease Mice

**DOI:** 10.3390/ijms231911560

**Published:** 2022-09-30

**Authors:** Peilin Sun, Hua Zhu, Xue Li, Weixiong Shi, Yaxi Guo, Xiaopeng Du, Ling Zhang, Lei Su, Chuan Qin

**Affiliations:** 1NHC Key Laboratory of Human Disease Comparative Medicine, Beijing Engineering Research Center for Experimental Animal Models of Human Critical Diseases, International Center for Technology and Innovation of Animal Model, Comparative Medicine Center, Institute of Laboratory Animal Sciences, Chinese Academy of Medical Sciences (CAMS) & Peking Union Medical College (PUMC), Beijing 100006, China; 2Changping National Laboratory (CPNL), Beijing 102206, China

**Keywords:** gut microbiota, metagenomics, metabolomes, Alzheimer’s disease

## Abstract

A common symptom in Alzheimer’s disease (AD) is cognitive decline, of which the potential pathogenesis remains unclear. In order to understand the mechanism of gut microbiota in AD, it is necessary to clarify the relationship between gut microbiota and metabolites. Behavioral tests, pathological examination, metagenomics, and metabolomics were applied to analyze the difference of gut microbiota and metabolome between APP^swe^/PS1^ΔE9^ (PAP) mice with cognitive decline and age-matched controls, and their possible correlations. Our results showed that PAP mice and health mice had different structures of the bacterial communities in the gut. The abundances and diversities of the bacterial communities in health mice were higher than in PAP mice by metagenomics analysis. The abundances of *Libanicoccus massiliensis*, *Paraprevotella clara*, and *Lactobacillus amylovorus* were significantly increased in PAP mice, while the abundances of *Turicibacter sanguinis*, *Dubosiella newyorkensis*, and *Prevotella oris* were greatly reduced. Furthermore, PAP mice possessed peculiar metabolic phenotypes in stool, serum, and hippocampus relative to WT mice, as is demonstrated by alterations in neurotransmitters metabolism, lipid metabolism, aromatic amino acids metabolism, energy metabolism, vitamin digestion and absorption, and bile metabolism. Microbiota–host metabolic correlation analysis suggests that abnormal metabolism in stool, serum, and hippocampus of PAP mice may be modulated by the gut microbiota, especially *T. sanguinis*, *D. newyorkensis*, and *P**. oris*. Therefore, abnormal metabolism activity is associated with gut microbiota in Alzheimer’s disease mice. Our results imply that modifying host metabolism through targeting gut microbiota may be a novel and viable strategy for the prevention and treatment of AD in the future.

## 1. Introduction

Alzheimer’s disease (AD) is a neurodegenerative disease characterized by progressive decline in cognitive function and loss of neurons and synapses. Females display a two-fold increase in the incidence of AD compared to males [1]. There are several theories about the pathogenesis of AD. The amyloid (Aβ) hypothesis has been dominant in the pathogenesis of AD, yet the majority of drugs developed based on this hypothesis have failed [2,3]. The possible reason for this is that these drugs have a single target, whereas AD is a chronic and complex disease involving multiple pathophysiological changes. Therefore, we need to revisit the pathogenesis of AD and demonstrate the association between different mechanism hypotheses in a holistic and systematic way. 

In recent years, an increasing number of clinical trials [4,5] and animal experiments [6,7] have found that the abundance and diversity of gut microbiota are altered in patients and animal models with AD, and that changes in gut flora are closely related to the pathogenesis of AD. Previous work from our lab found that the abundance and diversity of gut microbiota of 5-month-old PAP mice were significantly changed compared with the control group by 16S rRNA sequencing test, and the imbalance of gut microbiota may be closely related to the occurrence of cognitive impairment [8]. However, more reliable studies are still needed to examine the pattern of gut microbial composition of AD and to determine their potential role in promoting AD symptoms.

It is known that the intestinal flora as an important metabolic “organ” in the body, affecting the overall metabolism of the body, and when its community structure is changed, the physiological metabolism of the body will be changed accordingly [9]. Therefore, the metabolism of the body is influenced by both itself and the intestinal flora, and there is a “co-metabolic” process. Several human and animal studies have described differences in metabolic pathways and metabolites between AD and cognitively normal controls by metabolomics [10,11,12]. It has been shown that the pathogenesis of AD is associated with disturbances in metabolic pathways such as primary lipid metabolism, purine metabolism, amino acid metabolism, and oxidative phosphorylation [10]. Microbial-host co-metabolites, including serotonin, tryptophan catabolic products, bile acids, short chain fatty acids (SCFA), amino acid neurotransmitters, and catecholamines, may have key roles in mediating microbial effects on neurotransmission and disease development [13]. 

Although gut microbes may have an impact on the development of AD, the questions of exactly what these microbes are, what functions they have, and whether and to what extent these functions actually occur, are unclear. Therefore, in this study, we examined the structure and composition of the gut microbiota extracted from colonic contents of PAP mice and age-matched WT mice through metagenomic analysis. Additionally, metabolic profiles of feces, serum, and hippocampus were interrogated using liquid chromatography-mass spectrometry (LC/MS). Spearman correlation analysis was conducted for evaluating the association between microbes and metabolites. The primary purpose of this study was to identify changes in gut microbiome, as well as host metabolomes in stool, serum, and hippocampus, that are associated with AD, and ultimately to investigate the microbiota–host metabolic interaction.

## 2. Results

### 2.1. Evaluation of Learning and Memory Capability in PAP Mice

PAP mice demonstrated a lower percentage of spontaneous alternation in Y-maze, as compared to WT mice (Figure 1a). One week later, we used a well-established protocol of delayed Y maze for assessment of short-term spatial memory, and PAP mice showed reduced frequency of entering the novel arm, as well as less time staying in the novel arm, as compared to WT mice (Figure 1b,c). 

In Morris water maze (MWM) test, the escape latency of PAP mice was significantly longer than that of WT mice (Figure 1d), indicating that spatial learning may be impaired in PAP mice. Next, the spatial memory retention of mice was tested by the probe trial. The PAP mice swam randomly throughout the tank, whereas WT mice searched for the target quadrant preferentially as shown in Figure 1e. Furthermore, a significant decrease in both the frequency of platform crossings and time spent in the target quadrant was observed in PAP mice compared to WT mice (Figure 1f,g).

In comparison to WT mice, the PAP mice exhibited severe cognitive deficits, which is a key feature in AD patients.

### 2.2. Pathological Changes of Brain and Intestinal Tissues in PAP Mice

It is known that neuroinflammation involving microglia is an important driving factor of Alzheimer’s disease. Ionized calcium binding adapter molecule (Iba-1) is specifically expressed in microglia. Consistent with the robust inflammation found in AD patient brains and mice models, higher number of Iba1-positive cells was observed in PAP mice of 5-month age (Figure 2a). 

Intestinal barrier is a defense system that separates the intestinal cavity from the internal environment to prevent the invasion of intestinal antigens and dangerous components. The morphological changes of the ileal epithelium of PAP mice and WT mice compared by hematoxylin-eosin (HE) staining (Figure 2b). It was found that the ileal tissue structure of WT mice was normal, while the ileum of PAP mice showed edema of the mucosal layer and localized enlargement of the tissue gap (shown by black arrows), and the villi of the ileum atrophied and were loosely arranged. 

### 2.3. Metagenomic Sequencing Revealed Significant Differences of Gut Microbiota between PAP Mice and WT Mice

Intestinal contents extracted from the distal colons of WT and PAP mice. Through quality control and data filtering, high-throughput sequencing produced a total of 105Gb of clean metagenomic data, with an efficiency of more than 99.61% (Table 1). The rarefaction curves based on the Core-Pan genes gradually flattened, and as more data approached, it showed that the collected samples could meet the requirements of subsequent bioinformatics analysis (Appendix A). We performed gene level analysis by constructing non-redundant gene sets. As shown in Figure 3a, Venn plots showed 94,831 genes that are unique in WT mice and 68,830 genes unique in PAP mice. There were 856,900 genes in both groups (Figure 3a). Based on Bray–Curtis (Figure 3b,c), the Principal Coordinates Analysis (PCoA) and Non-Metric Multi-Dimensional Scaling (NMDS) were established to study the similarity between the two groups of microbial communities. The analysis shows that the microbiota composition of the PAP mice has great heterogeneity, which is a significantly different situation from that of WT mice, a result that is consistent with previous studies. At the phylum level, we found that the microbial structure of colonic contents of PAP mice was significantly differed from that of WT mice (Figure 3d), which was manifested by a decrease in Bacteroidetes and an increase in Firmicutes. In addition, the Firmicutes/Bacteroidetes ratio was significantly increased in PAP mice compared with WT mice (Figure 3e). According to the general situation of genus level bacterial communities, 281 genera in the two groups showed different relative abundances (Appendix A). Hierarchical heatmap showed that the 35 key genera detected in all samples exhibited different patterns between the PAP and WT groups (Figure 3f). We then corrected multiple comparisons to show significant species differences between PAP and WT mice (Figure 3g). The species *Libanicoccus massiliensis*, *Paraprevotella clara*, and *Lactobacillus amylovorus* were more abundant in PAP mice, while species *Turicibacter sanguinis*, *Dubosiella newyorkensis*, *Prevotella oris*, *Alistipes timonensis*, and *Neglecta timonensis* in WT mice. 

To describe the potential relationship among different gut microbial communities, we constructed co-occurrence networks of species from each group based on significant Spearman correlations. The co-occurrence network of WT group and PAP group were composed of dispersed genera belonging to four main phyla (Firmicutes, Bacteroidetes, Proteobacteria, and Actinobacteria) (Figure 4a,b). The WT group showed a highly positively correlated co-occurrence networks among species (Figure 4a). As shown in Figure 4b, the microbial community of PAP mice has a more complex network. The correlation between Firmicutes was significantly decreased, and the correlation between Bacteroidetes and Proteobacteria was increased. To sum up, the above analyses shows that the microbial relationship of PAP mice has changed compared with that of WT mice, further indicating that there is dysbiosis in the intestinal microecology of PAP mice.

### 2.4. Functional Analysis of Metagenomic Sequencing Revealed Disrupted Bacteria Functions in PAP Mice

Metagenomic sequencing data were used for functional analysis. To compare functional bacteria genes, we used the KEGG and CAZy databases based on clean data to compare the intestinal microbial functions of the two groups in the study cohort. A total of 4649 KEGG orthologous (KO) categories were identified by metagenomic analysis, including 6 KEGG Level 1, 45 KEGG Level 2, and 381 KEGG Level 3. At the first level of KEGG classification, metabolism, genetic information processing, and environmental information processing were the three main functions of the two groups of gut microbiota (Appendix A). At the second taxonomic level, carbohydrate metabolism and amino acid metabolism dominated gut microbiome function in WT mice and PAP mice (Appendix A). At the third taxonomic level, the most abundant functional pathways in these two groups were ko00230 (purine metabolism) and ko03010 (Ribosome) (Appendix A). 

These unique genes identified 523 enzyme commissions (EC) in the gut microbiota, which after confirmation in the CAZy database, consisted of 6 CAZy modules and 257 CAZy families. At the first taxonomic level of CAZy, glycoside hydrolases, glycosyl transferases, and carbohydrate-binding modules were the dominating three enzyme families in the gut microbiome of these two species (Appendix A). At the second taxonomic level, GH43, GT2, and GH2 were the most abundant enzyme families in the gut microbiome of WT mice and PAP mice (Appendix A). At the EC level, alpha-L-arabinofuranosidase (EC 3.2.1.55) was the most abundant enzyme in both group (Appendix A).

We calculated the Bray–Curtis distance of each classification level gene in the KEGG and CAZy database, and then plotted the PCoA map. At each taxonomic level, PCoA plots showed that functional components of the gut microbiota were isolated between WT and PAP mice (Appendix A). We performed ANOSIM analysis based on functional abundance at each taxonomic level. There was no significant difference between the two groups (*p* > 0.05). Therefore, we used a MetaStats analysis and LDA analysis to determine all significant differences in functional pathways and enzymes.

The MetaStats analysis showed that all the KEGG pathways based on level 1, level 2, and KO were disrupted in PAP mice, relative to WT mice (Figure 5a–c). For instance, the metabolic pathway activity of energy, lipids, vitamins, glycan, xenobiotics biodegradation, and nucleotides were all higher in the PAP mice. The unique gene number of phosphoglycolate phosphatase (K01091), ribosomal protection tetracycline resistance protein (K18220), and maltose O-acetyltransferase (K00661) were significantly more abundant in the PAP mice than in the WT mice, while 2-dehydro-3-deoxygluconokinase (K00874) and arabinan endo-1,5-alpha-L-arabinosidase (K06113) were lower (*q* < 0.05) (Figure 4c, Appendix A). We identified significantly different enzymes in the two groups by LDA (LDA score > 3, *p* < 0.05). The dates showed that the PAP mice had more enriched enzyme families for GT2 and GH20 compared to the WT mice (Figure 5d). 

### 2.5. Metabolomics Analysis Revealed Aberrant Metabolic Patterns in PAP Mice

Microbiota-derived metabolites influence the host through multiple pathways. Many metabolic products of gut microbiota may enter the bloodstream and exert important influences on the physiology and behavior of the hosts. To verify this assumption, we further performed nontargeted metabolomics through liquid chromatography-mass spectrometry (LC/MS) to determine whether or which metabolisms modulated by the gut microbiome were paralleled by an altered MGB axis. The fecal, serum, and hippocampal samples obtained from distinct groups were largely separated according to the Partial Least Squares Discriminant Analysis (PLS-DA) (Figure 6a–c), suggesting the dissimilar metabolic modes. The potential biomarkers were selected with the VIP value (>1) and statistical tests (*p* < 0.05) and were displayed in a volcano plot in Figure 6d–f. 

Pathway analysis for the potential biomarkers was conducted via the KEGG topology analysis (Figure 7). Eight metabolic pathways were enriched between the PAP mice and WT mice in the stool (Figure 7a), including the vitamin digestion and absorption, synaptic vesicle cycle, gastric acid secretion, neuroactive ligand-receptor interaction, biotin metabolism, pyrimidine metabolism, purine metabolism, and bile secretion. Steroid hormone biosynthesis, ovarian steroidogenesis, prostate cancer, citrate cycle (TCA cycle), glyoxylate and dicarboxylate metabolism, endocrine resistance, prolactin signaling pathway, tyrosine metabolism, butanoate metabolism, and dopaminergic synapse were screened out as the metabolic pathways between the two groups in the serum (Figure 7b). Furthermore, vitamin digestion and absorption, nicotinate and nicotinamide metabolism, glycerophospholipid metabolism, and biosynthesis of unsaturated fatty acids were screened out as the metabolic pathways between the two groups in the hippocampus (Figure 7c). 

Among them, the differentially expressed metabolites with definite information were identified between the two groups (Appendix A). 23-nordeoxycholic acid, 23-norcholic acid, 7-ketodeoxycholic acid, deoxycholic acid, β-muricholic acid, and cholic acid, which are involved in bile acid metabolism, were significantly reduced in the blood of PAP mice. Glycolithocholic acid was significantly decreased in the feces of PAP mice, while dehydrocholic acid was increased. Involved in vitamin absorption and metabolic pathways included cholecalciferol and pyridoxal were reduced in both feces and hippocampus of PAP mice, while biocytin was increased in feces, and flavin adenine dinucleotide and riboflavin were reduced in blood and hippocampus of PAP mice, respectively. Involved in the biosynthesis of unsaturated fatty acids such as arachidic acid is increased in the blood of PAP mice, and eicosapentaenoic acid, neuronic acid and erucic acid are all higher expressed in the hippocampus of PAP mice. Alpha-Ketoglutaric acid and acetoacetate as the members of the butanoate metabolism pathway is significantly reduced in the blood of PAP mice, whereas fumaric acid was significantly increased. In addition to butanoate metabolism, fumaric acid was also involved in various other metabolic pathways, including citrate cycle, nicotinate and nicotinamide metabolism, alanine, aspartate and glutamate metabolism, and tyrosine metabolism. Participated in the biosynthesis of steroid hormones including progesterone, estradiol, and testosterone were reduced in the blood of PAP mice; tetrahydrocortisone was reduced in their hippocampus, while aldosterone and desoxycortone were increased in the feces and blood of PAP mice, respectively. 

In general, AD is characterized by abnormal level of neurotransmitters such as serotonin, dopamine, γ-aminobutyric acid (GABA) and indole. The gut microbiota may be involved in the synthesis and metabolism of neurotransmitters, so we further investigated whether the difference of metabolites between the two groups is involved in the metabolic process of neurotransmitters. As shown in Figure 8, many differential molecules were found, including levodopa, pyroglutamic acid, glutamine, indole-3-acetic acid, 3-(2-hydroxyethyl) indole, histamine, and acetylcholine, which are related to the neurotransmitter metabolic network such as phenylalanine and tyrosine metabolism, tryptophan metabolism, histidine, aspartate metabolism, and glutamate metabolism. In addition, metabolites involved in purine metabolism showed levels in AD. In summary, the analysis further implicated that gut microbes might be involved in the metabolism of neurotransmitters.

### 2.6. Microbiota–Host Metabolic Interaction

To explore the relationship between gut microbiota and metabolites, a correlation matrix assessment was conducted using Spearman’s correlation analysis based on 8 bacterial species and 58 metabolites that significant altered between PAP and WT mice, which is presented as a heatmap shown in Figure 9. *D**. newyorkensis* has a negative relationship with eicosapentaenoic acid and arachidic acid belongs to the long chain fatty acids in serum. Whereas it was positively correlated with metabolites including kynurenic acid, 5-hydroxyindole-2-carboxylic acid, glutathione, and capriloylglycine, which were involved in tryptophan metabolism, glutamate metabolism, glycine metabolism, and glycospholipid metabolism pathways in hippocampus. However, comparing with *D**. newyorkensis*, *L**. massiliensis* and *L**. amylovorus* were found to have opposite correlation with the above-mentioned metabolites. There was a significant negative correlation between *L**. amylovorus* and metabolites involved in bile acid metabolism of serum, including cholic acid, deoxycholic acid, 7-ketodeoxycholic acid and 23-norcholic acid. *T**. sanguinis*, *P**. oris*, and *N**. timonensis* are positively correlated with indole derivatives of feces, such as indole-3-acetic acid and 3-(2-hydroxy) indole. They were also positively correlated with medium-chain fatty acids (sebacic acid, azelaic acid and suberic acid) in feces. 

## 3. Discussion

Cognitive dysfunction is one of the most common symptoms in AD. In this study, we found that the learning and memory ability of PAP mice were impaired, which was consistent with our previous findings [8]. Furthermore, the expression of Iba-1 in the hippocampus of PAP mice was significantly higher than that of WT mice, indicating that neuroinflammation occurred in PAP mice. Microglia are resident immune cells of the central nervous system. The activation of central glial cells plays a key role in the neuroinflammatory response in AD. Neuroinflammation is closely related to neuropsychiatric disorders such as memory disorder, brain injury, and depression [14]. The change of gut microbiota will lead to Inflammatory reaction, and the activation of immune system will affect brain function through glial cell activation and cytokine release, thus causing brain activity disorder [15]. Bacterial metabolites and their activated inflammatory factors can also cause intestinal mucosal damage and increase intestinal permeability. *L. reuteri*, for example, causes longer villi in the ileum of chickens [16]. Our results showed edema of ileal mucosa and atrophy of ileal villi in PAP mice, indicating that the composition of the gut microbiota may indeed affect intestinal morphology. Interestingly, compared with WT mice, PAP mice do have a peculiar gut microbial pattern. For example, the observed species of the gut microbiota in PAP mice were significantly reduced, which was also reported by Zhang et al. [7]. Another feature found in this research is that PAP mice have a higher proportion of Firmicutes and a lower proportion of Bacteroidetes than WT mice. Analogous studies reported similar changes in microbiota, in which the relative abundance of Bacteroidetes in older mice is lower than that in young mice, and the relative abundance of Firmicutes in older mice is higher. The proportion of Firmicutes/Bacteroidetes increased several fold with aging [17,18]. In addition, our results showed that PAP mice had less abundance of genera including *Turicibacter*, *Neglecta*, and *Dubosiella*. In AD patients, it has been reported that the abundance of Turicibacter and *Dubosiella* were decreased, and there was a relationship between the decreased abundance of Turicibacter and the increased level of YKL-40 in CSF [19,20]. Most previous studies have used 16S rRNA to determine the composition of the intestinal microbial community in Alzheimer’s disease. Many sequences obtained by 16S rRNA sequencing have been annotated to the genus or family level [21]. On the basis of 16S rRNA sequencing analysis, we can also conduct in-depth research on the genes and functions of gut microbiota through metagenome sequencing, and identify microorganisms at the species level. Our study showed that, compared with WT mice, the relative abundances of *L**. massiliensis*, *P**. clara*, and *L**. amylovorus* in PAP mice were significantly increased, a while the relative abundances of *T**. sanguinis*, *D**. newyorkensis*, *A**. timonensis*, *P**. oris*, and *N**. timonensis* were opposite. *Prevotella* is the predominant bacteria in the human gut [22]. Recent studies have shown that polysaccharides can be used by *P**. copri* to produce succinic acid, which has been reported to enhance the immune response [23]. However, there are few reports on *P**. oris*, and its role needs further study. It is worth noting that *D. newyorkensis* is used as a patented probiotic to regulate weight loss and prevent metabolic and immune diseases such as obesity, diabetes, metabolic syndrome, and abnormal lipid metabolism [24]. *L**. amylovorus* is one of the first discovered strains and has been widely studied. *Lactobacillus amylovorus* plays a beneficial role in anti-inflammatory and promoting intestinal health [25]. The results showed that the abundance of *L**. amylovorus* in the PAP group was higher than that in the WT group, which provided further evidence for the adverse reaction of *L**. amylovorus*. However, the disturbance of these species is rarely reported in AD, and whether there is a causal relationship between them and cognitive decline in AD needs further study.

Metabolic characteristics are unique in individuals, and changes in metabolite concentrations contribute to understanding the state of disease and its underlying pathophysiological mechanisms. In AD, metabolic disorders have been observed in many tissues and biological fluids, involving many organs and systems besides the central nervous system [26]. Metabolomic analysis found that many metabolic pathways and reactions in AD were significantly disturbed, including lipid homeostasis, fatty acid biosynthesis, membrane lipid remodeling, methionine/arginine/glutamate/polyamine metabolism, mitochondrial bioenergetics, production of both reactive oxygen and nitrogen species (ROS and RNS), biosynthesis of neurotransmitter, synaptic transmission, calcium homeostasis, inflammatory/immune response, and apoptosis. It is generally believed that impaired lipid homeostasis and biosynthesis of neurotransmitter are the most frequently misregulated molecular pathways in AD pathophysiology [27]. Likewise, PAP mice also had abnormal lipid metabolism, neurotransmitter metabolism, and fatty acid biosynthesis compared with WT mice in our metabolomics results. Lysophosphatidylcholines (lysoPCs) are phosphatidylcholines (PCs) products that maintain the normal integrity of cell membranes and are also important cell signaling molecules. LysoPCs are the preferred carriers of polyunsaturated fatty acids (PUFAs) entering the brain through the blood-brain barrier (BBB). Regarding phospholipids, a recent study showed that plasma concentrations of three well-defined PCs (PC 16:0/20:5, PC 16:0/22:6, and PC 18:0/22:6) were decreased in AD and MCI patients compared with age-matched controls [28]. This decline is also associated with poor cognitive ability in normal elderly people, which indicates that there may be an association between phospholipid homeostasis disorder and cognitive function [29]. In the present study, thirteen lysoPCs and eight PCs were identified as differential metabolites and decreased in the serum of PAP mice. The deficiency of circulating lysoPCs pools containing long-chain fatty acids may limit the amount of long-chain fatty acids supplied to the brain, including PUFAs such as DHA, and play a role in the pathobiology of AD. The differential metabolites between PAP mice and WT, including serotonin, dopamine, histidine, and GABA were involved in a neurotransmitter metabolic network. GABA is considered to be the main inhibitory neurotransmitter. Recent studies have shown that AD shows a change pattern of GABA metabolites in feces, which may be related to gut dysbiosis [30]. Similarly, changes in GABA precursors in AD, such as N-acetyl-l-glutamine, glutamine, and pyroglutamic acid have also been observed in this study. More and more studies have reported abnormalities dopamine (DOPA) signals in AD. Currently, there is no coherent dopamine hypothesis linking neurobiology to behavior in AD [31]. Current analysis of metabolites indicates that amino acid metabolism is abnormal, such as significant changes in several tyrosine and phenylalanine derivatives related to DOPA synthesis and metabolism. Recent studies on human and animal models have shown that the disorder of cholinergic system may be the basis of AD related behavioral symptoms [32]. Some studies have shown that the elevation of acetylcholine (ACh) in AD patients is significantly correlated with the improvement of symptoms of AD and the decreases of ACh [33]. In this study, the level of ACh in AD decreased, suggesting that the intestinal microbes in AD patients may regulate ACh through the gut brain microbial axis, and ultimately lead to behavioral defects in AD. In addition, compared with wild-type mice, PAP mice have significantly impaired amino acid metabolism, such as aromatic amino acids (phenylalanine, tyrosine, and tryptophan), glutamate, histidine, aspartate, glycine, and alanine. This finding may also explain the decline in cognitive function in AD, because amino acids play an important role in neurotransmitter synthesis, protein biosynthesis and energy remodeling. In the study of nervous system diseases, amino acids are always important. For example, elevated plasma alanine levels have been identified as a therapeutic indicator for schizophrenia [34]. Alanine supplementation can improve cognitive function in patients with schizophrenia [35]. There is no doubt that energy metabolism plays an important role in maintaining the normal life of mammals. Similarly, we found that the serum and hippocampal energy metabolism of PAP mice were significantly lower than that of WT mice, such as TCA cycle intermediates, citric acid, fumaric acid, malic acid, and alpha-Ketoglutaric acid. In this study, the concentration of NAD^+^ and FAD in serum and hippocampus of PAP mice was also significantly lower than that of WT mice. It is well known that energy metabolism plays a key role in cognitive function [36]. Therefore, we speculate that the decrease of energy metabolism may be the cause of cognitive impairment in AD. Metabolomics elucidates the disordered metabolic process of AD, which is the result or cause of ad pathophysiology. In conclusion, exploratory metabolomic studies on stool, serum, and hippocampus of PAP mice show that the disordered metabolic process includes but is not limited to lipid homeostasis, neurotransmitter biosynthesis, amino acid metabolism, and energy metabolism, which are the basis of AD.

In order to understand the role of gut microbiota in AD metabolism, it is essential to analyze the relationship between metagenomics and metabolomics. In the present study, *P. oris* and *T. sanguinis* were significantly reduced, which means that the decrease in GABA precursor levels may be due to the reduction of *P. oris* and *T. sanguinis*. Therefore, it may be promising to further study the specific microorganisms related to the neural circuits related to the GABA receptor-mediated function in the pathophysiologically related areas of AD. There is increasing evidence that the presence of intestinal SCFA in the diet and the production of SCFA by gut opportunistic bacteria after food carbohydrate fermentation may be environmental triggers of AD [37]. Recent analysis of fecal SCFA showed that the increase in SCFA levels was associated with Firmicutes, but decreased in AD [38]. Similarly, we found that the SCFA (such as propionic acid and butyric acid), with its antibacterial, anti-inflammatory, and immunomodulatory effects, was lower in PAP mice than in WT mice. Previous studies on fecal SCFA in AD are consistent, and it can be assumed that fecal SCFA can indicate intestinal mucosal SCFA levels [38]. Studies have shown that tryptophan can be directly metabolized by intestinal microorganisms such as *Lactobacillus*, and its metabolites include indoleacrylic acid (IA), indolepropionic acid (IPA), and indoleacetic acid (IAA) [39]. Our results showed that there was a significant positive correlation between *T**. sanguinis* and IAA (Figure 9), and the abundance of *T**. sanguinis* was reduced in PAP mice. It is suggested that the decreased level of IAA in AD may be caused by *T**. sanguinis*. In addition, the abundance of *L. massiliensis* and *T. sanguinis* were closely related to these differential metabolites involved in DOPA synthesis and metabolism, further suggesting that these two species may play a potential role in DOPA abnormal signal transduction in AD by regulating amino acid metabolism. Therefore, the association analysis based on metagenomics and metabolomics lays a foundation for revealing the potential molecular mechanism of AD mediated by gut microbial disorders, screening potential biomarkers, and more accurate diagnosis and treatment.

## 4. Conclusions

Overall, our results suggest that the impaired learning and memory ability of PAP mice is related to the disturbance of gut microbiota and alteration of metabolites in fecal, serum, and hippocampus. As shown in summary of Figure 10, the abundance and diversity of bacterial communities in PAP mice were significantly lower than that in WT mice. The abundance of *L. massiliensis*, *P. clara*, and *L. amylovorus* increased significantly in PAP mice, while the abundance of *T**. sanguinis*, *D**. newyorkensis*, and *P**. oris* decreased significantly in PAP mice. There were several metabolic pathways that were significantly altered between PAP and WT mice, such as neurotransmitters metabolism, lipid metabolism, aromatic amino acids metabolism, energy metabolism, vitamin digestion and absorption, and bile metabolism. Additionally, microbiota–host metabolic correlation analysis indicates that these metabolic changes may be attributed to the gut microbiota, especially *T**. sanguinis*, *D**. newyorkensis*, and *P**. oris*. In particular, the analysis of the interaction between gut microbiota and metabolites can provide clues to further understand the mechanism of cognitive deterioration in AD, and clarify whether the origin of these changes is related to gut microbiota. Abnormal microbial metabolites may play an important role in the pathogenesis of AD, and further studies are needed to confirm the relationship between gut-microbiome-mediated metabolites and the central nervous system. This study provided insights into the relationship between gut microbiota and metabolism in AD, and provided a possible future model for AD intervention targeting specific metabolism related microorganisms.

## 5. Materials and Methods

### 5.1. Animals

Twenty female mice (4 months old, weighing 25–30 g) were purchased from Huafukang Biotechnology Co. Ltd. (Beijing, China), including 10 APP^swe^/PS1^ΔE9^ (PAP) transgenic mice and 10 littermate wild-type control (WT) mice. All mice were placed in a specific pathogen-free (SPF) facility of the Institute of Laboratory Animal Science, Chinese Academy of Medical Sciences, where they could obtain standard food and water free of charge. Fully controlled feeding conditions: temperature: 22 ± 1 °C; humidity: 55 ± 5%; light: 12-h light/dark cycle and lights on at 5:00. Protocols for all animal studies were compliant with and approved by the Institutional Guidelines for the Care and Use of Laboratory Animals, Institute of Zoology (Beijing, China). 

### 5.2. Behavioral Tests

#### 5.2.1. Y Maze

Behavioral tests were performed by three arms of equal length (30 × 7 × 15 cm). In order to record the activity of the animals, the camera was fixed on the ceiling 100 cm above the center of the maze. According to the above method, the continuous spontaneous change test was used to measure spatial working memory. Each mouse was placed at the end of one arm and was allowed to explore the maze freely for 5 min. Record the order in which the mouse enters the arm to calculate the percentage change. Input is recorded only when all mouse limbs are on the arm. The percentage of change was calculated by continuously dividing the number of probes in the three arms by the total number of possible changes and multiplying by 100. According to the above method, the delayed spontaneous change test was conducted one week after the end of the continuous spontaneous change test. During the training phase, mice explored two of the arms for 5 min while closing the new arms. As part of the test phase, the subjects were placed in the start arm, held in cages for 15 min, and then allowed to explore all three arms for five minutes. During the test phase, record the time spent on the new arm (stay) and the number of inputs performed on the new arm. Dwell time and entry time were indicators of inspective and inquisitive behavior, respectively.

#### 5.2.2. MWM Test

Spatial learning and memory were assessed using the MWM test according to standard procedures [40]. The water maze consists of a circular pool with a diameter of 100 cm and a height of 50 cm. It is divided into four equal quadrants by two intersecting parallel image lines. In one quadrant of the swimming pool, the platform is submerged 1 cm below the water surface. The escape latency refers to the time needed for the mouse to reach and climb the escape platform. When the mice did not reach the platform within 60 s, the experimenters guided them to the platform and recorded the escape latency within 60 s. In both cases, the mice were allowed to rest on the platform for 15 s and then returned to the cage. Mice were tested 3 times a day for 5 consecutive days. During the space probe test (day 6), the platform was removed. Mice were released from the quadrant opposite the target quadrant and allowed to swim freely for 60 s. The swimming route and crossing times in one minute were recorded. After the probe test on the 7th day, the visible platform test was carried out, the platform was raised to the water surface, and each mouse was tested. All experiments were performed at the same time every day. 

### 5.3. Morphological Examination

The intestinal tissues of mice were immersed in 10% neutral formalin buffer and embedded in paraffin. After dewaxing, sections were cut with a 4 μm thick microtome, washed with PBS, stained with H&E staining and subjected to routine immunohistochemistry. In summary, mouse cerebral hemispheres were fixed in 10% formalin solution, cut into 5-µm-thick concave sections, stained with Iba-1 antibody (1:1000, #17198, Cell Signaling Technology), and overnight at 4 °C. Then, after incubation with the second antibody (HRP-labeled anti-mouse IgG), the immune response was observed and rinsed three times in PBS, followed by DAB (ZSGB-BIO, Beijing, China) [41].

### 5.4. Sample Collection and Preparation

In this study, fresh stool was collected on the last 3 days of MWM test, and blood was collected within 24 h after MWM test. Then the mice were injected with cold sterile saline through the ascending aorta to collect brain and intestinal specimens. Blood was collected and centrifuged at 3500× g for 15 min at 4 °C. The supernatant was removed and stored at −80 °C for future use. Stool and left hippocampus were frozen in liquid nitrogen until analysis. The right hemisphere was fixed with 4% paraformaldehyde for histopathological analysis.

### 5.5. Metagenomic Analysis

The contents in the colon of mice of WT and PAP were collected under SPF conditions and maintained at −80 °C before use. Each sample was prepared with approximately 1 μg of DNA. A NEBNext^®^ Ultra^™^ DNA Library Prep Kit for Illumina (NEB, USA) was used to prepare sequencing libraries. Sequencing of the libraries was performed at Novogene Bioinformatics Technology Co., Ltd. (Tianjin, China) on the Illumina Hiseq X platform (insert size 350 bp, read length 150 bp). Sequences with low quality were discarded, and high-quality sequences were assemble with SOAPdenovo version 2.04 (http://soap.genomics.org.cn/soapde novo.html, accessed on 24 August 2021) [42]. Meta GeneMark version 2.10 (http://topaz.gatech.edu/GeneMark/, accessed on 26 August 2021) was used to predict the genes. A non-redundant gene catalog was produced by removing redundant genes (99% identity, 90% overlap) from CD-HIT version 4.5.8 (http://www.bioinformatics.org/cd-hit, accessed on 27 August 2021) [43]. DIAMOND version 0.9.9 (https://github.com/bbuchfink/diamond/, accessed on 30 August 2021) was used to align reads for taxonomy functional assignment and taxonomic identity [44]. The LCA algorithm from MEGAN software (Version 6.12, Tübingen, Baden-Württemberg, Germany) system was used for annotations [45]. DIAMOND Version 0.9.9 was used to assign predicted unigenes to the Kyoto Encyclopedia of Genes and Genomes (KEGG), evolutionary genealogy of genes: carbohydrate-active enzymes database (CAZy) [46]. The abundances of functional annotations at each level were the sum of their abundances. To understand the correlation between different species, we established a co-occurrence network based on metagenomic data. Using Spearman’s correlation analysis, the co-occurrence network was constructed in the WT and PAP samples, respectively, according to relative abundance of each species. A visualization of the significant correlated species (false discovery rate < 0.05, rho ≥ 0.25) was made using Cytoscape version 3.6.1 (http://www.cytoscape.org, accessed on 10 September 2021) [47].

### 5.6. Metabonomic Analysis Based on Liquid Chromatography-Mass Spectrometry (LC/MS)

The feces, serum, and hippocampal samples were subjected to metabolomics analysis using LC/MS technique (Novogene Bioinformatics Technology Co., Ltd. Tianjin, China). Each 100 mg sample were mixed with 500 μL 80% methanol aqueous solution, vortexed for 30 s, incubation 5 min at 4 °C. Following centrifugation at 15,000× *g* rpm for 20 min at 4 °C, the supernatant was transferred to LC/MS [48]. Analysis of the LC/MS data was performed in accordance with the previous study [49]. Formic acid (0.1%) and ammonium acetate (5 mM) were used as solvent A for positive and negative experiments, respectively. Methanol was the solvent B. The gradient elution of solvent B was performed as follows: 2% for 0–1.5 min; 85% for 3 min; 100% for 10 min; 2% for 11 min; 2% for 12 min. The spraying voltage was 3.8 kv and the capillary temperature is 320 °C. Masses between 100 and 1500 *m*/*z* were obtained. Process the raw data using CD 3.1 library search software. Data processing include peak intensity, mass-to-charge ration (*m*/*z*), and retention time (RT). Based on additive ions, molecular ion peaks, and fragment ions, the normalized data was used to estimate the molecular formula. To obtain accurate qualitative and quantitative results, peaks were matched with the mzCloud (https://www.mzcloud.org/, accessed on 2 December 2021), mzVault, and MassList. Partial least squares discriminant analysis (PLS-DA) was used to assess differences in metabolic profiles between WT and PAP mice [50]. KEGG database was used to import all observed and predicted compounds. Important metabolite predictor (VIP) > 1, *p* value (*t*-test) < 0.05. Then important metabolites with changes > 1.5 times were selected for subsequent analysis.

### 5.7. Data Analysis and Statistics

SPSS 23.0 statistical software (New York, NY, USA) was used for multivariate logistic regression analysis (*p* < 0.05). Analyzing the differences in abundance between genera and metabolites was carried out using parametric and non-parametric tests, including Wilcoxon rank sum tests and *t*-tests. MetaX software (Version 2.68, Shenzhen, China) was used to conduct principal component analysis (PCA), fold change analysis, and partial least squares discriminant analysis (PLS-DA) [51]. The correlation between gut microbiota and metabolites in AD was determined using Pearson correlation coefficient. Unless otherwise stated, an adjusted *p* value < 0.05 should be considered statistically significant.

## Figures and Tables

**Figure 1 ijms-23-11560-f001:**
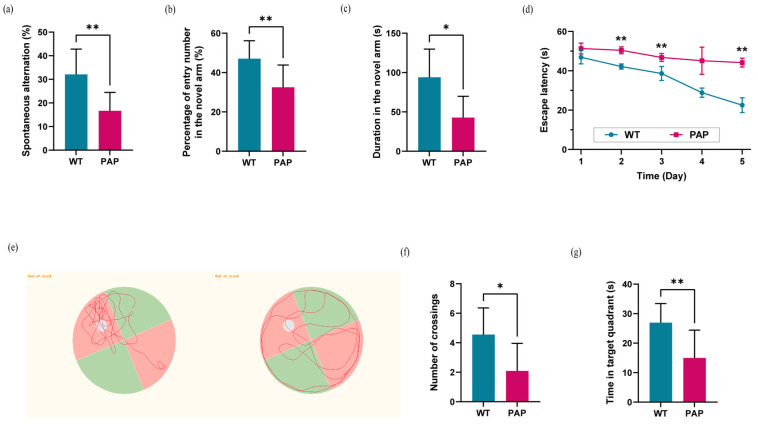
Evaluation of learning and memory ability evaluation in WT and PAP mice. (**a**) Spontaneous alternation in arm entries; (**b**) Frequency and (**c**) Time in the novel arm in Y maze test. (**d**) Mean escape latency in the hidden platform task; (**e**) Swimming paths, (**f**) frequency of crossings, and (**g**) time in the target quadrant in the probe trial. Results were displayed as mean ± SEM (*n* = 10 per group). Significant level: * *p* < 0.05; ** *p* < 0.01.

**Figure 2 ijms-23-11560-f002:**
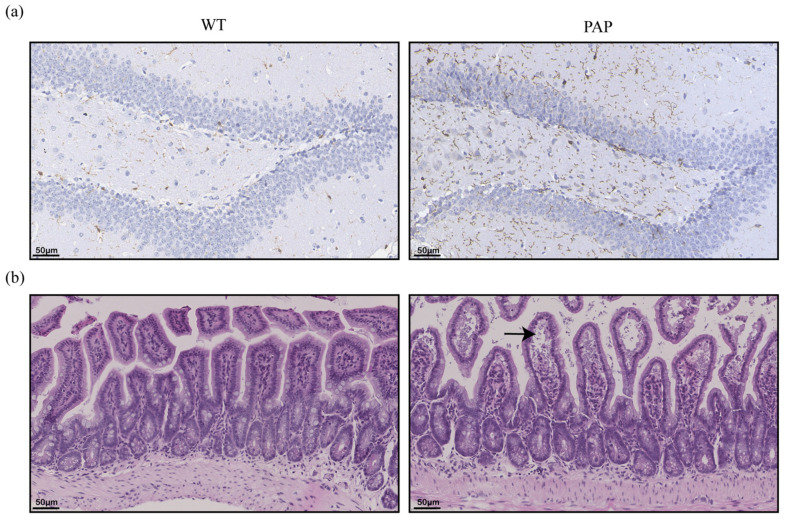
Pathological changes of brain and intestinal tissues in PAP mice. (**a**) Immunohistochemistry of Iba-1 expression in the hippocampus (scale bar: 50 µm); (**b**) HE staining of ileal tissue (black arrow head indicate swelled intestinal villi and enlarged tissue gap).

**Figure 3 ijms-23-11560-f003:**
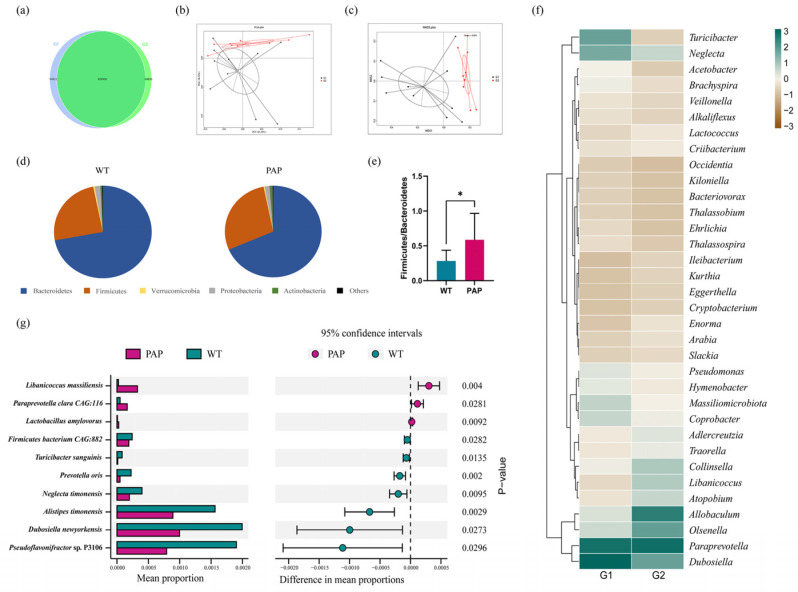
The shift of gut microbiota in WT and PAP mice according to the metagenomics data. (**a**) Venn diagram of the number of genes in WT and PAP. (**b**) Principal coordinate analysis (PCoA) and (**c**) Non-Metric Multi-Dimensional Scaling (NMDS) of the microbiota based on Bray–Curtis. (**d**) Gut microbiota composition at the phylum level and (**e**) Firmicutes/Bacteroidetes ratio in PAP and WT mice. (**f**) Heatmap showed relative abundance of the top 35 genera across two groups. (**g**) Changes in the gut microbiota at the species level selected from stamp analysis. * *p* < 0.05 by student’s *t*-test. G1 = WT group, G2 = PAP group.

**Figure 4 ijms-23-11560-f004:**
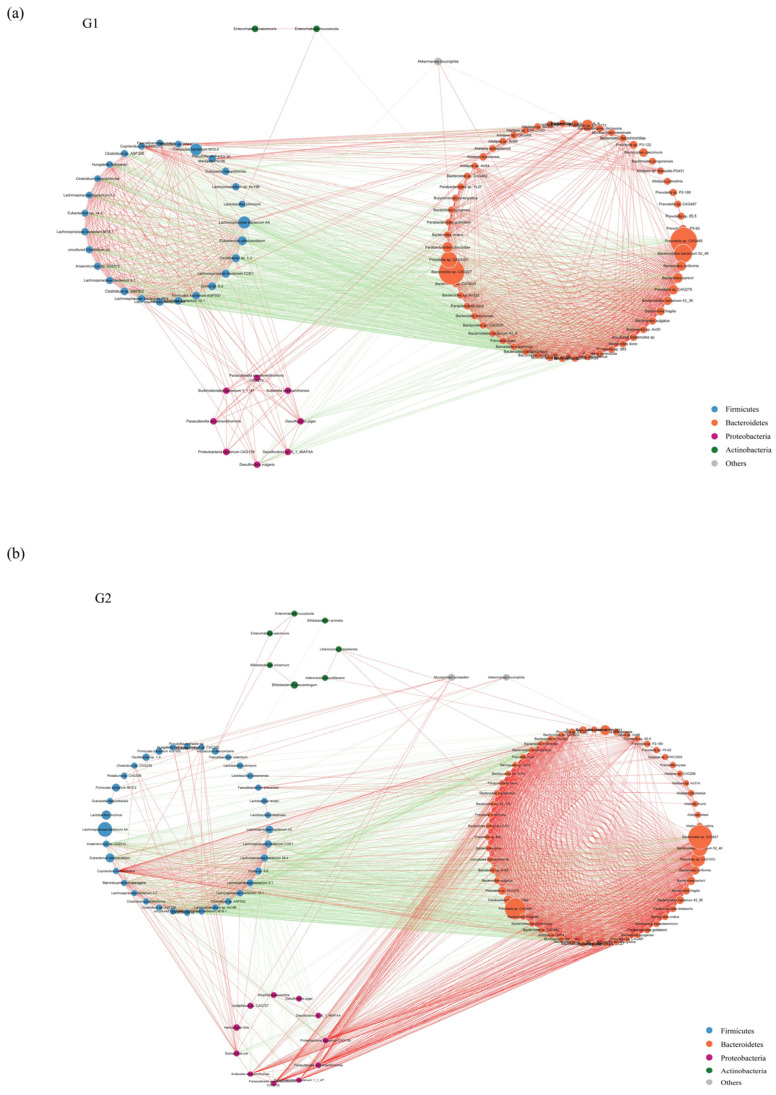
Species co-occurrence network between WT (**a**) and PAP (**b**) mice based on the Spearman correlation algorithms. Each node presents a bacterial genus. The node size indicates the relative abundance of each species per group, and the density of the dashed line represents the Spearman coefficient. Red links stand for positive interactions between nodes, and green links stand for negative interactions. G1 = WT group, G2 = PAP group.

**Figure 5 ijms-23-11560-f005:**
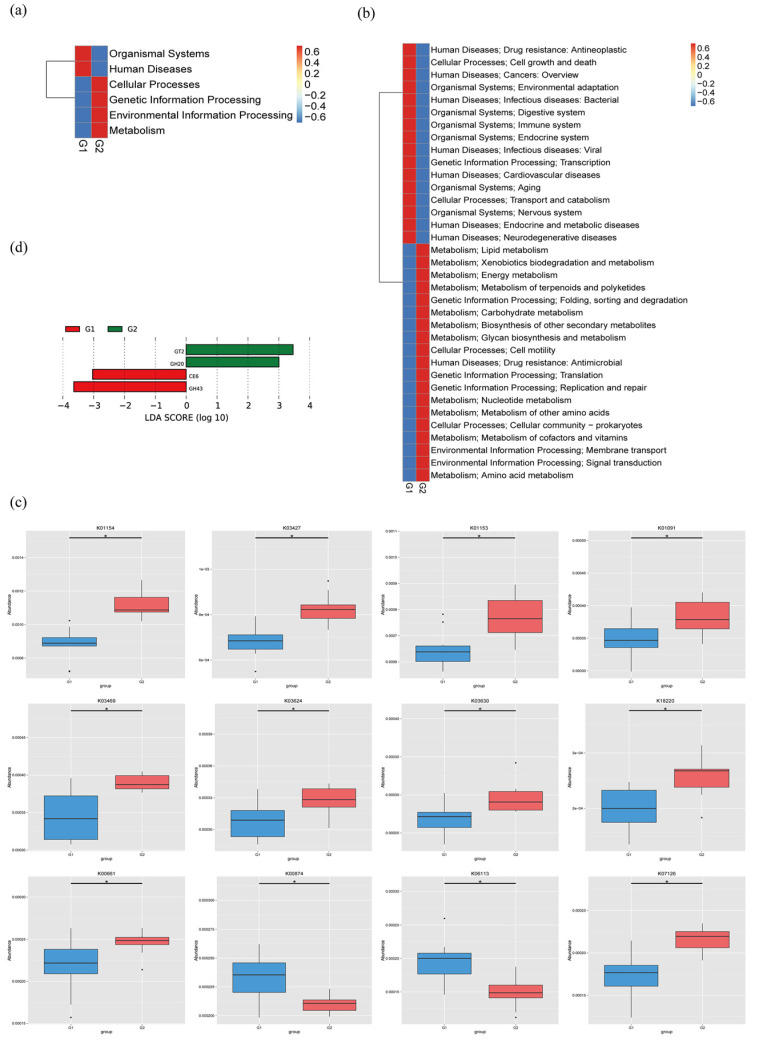
The average abundance of KEGG pathway differentially enriched in WT and PAP according to level 1 (**a**), level 2 (**b**), and KO (**c**). (**d**) LDA plot of the enriched enzymes abundant in microbiome from two groups. G1 = WT group, G2 = PAP group.

**Figure 6 ijms-23-11560-f006:**
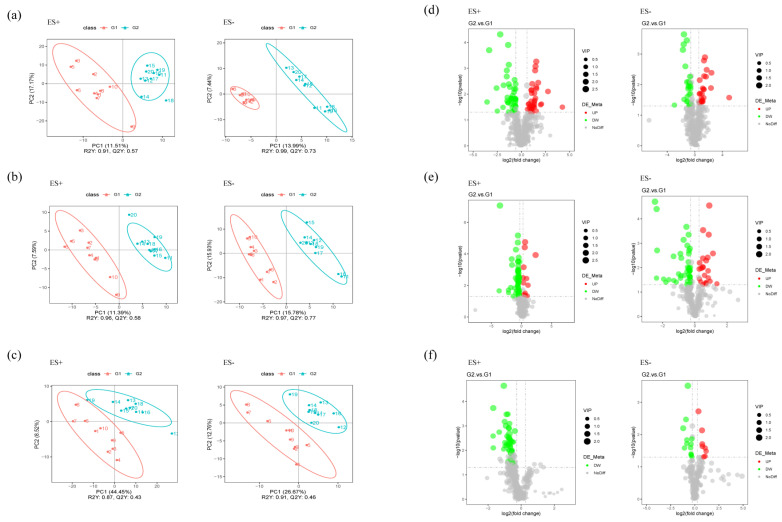
Multivariate statistical analysis of untargeted metabolomics data in the samples (positive ion combined with negative ion). (**a**–**c**) PLS-DA score plot of metabolomics data for each pairwise group of feces, serum, and hippocampus, respectively. (**d**–**f**) Volcano plots for the model-separated metabolites according to the conditions of VIP > 1 and *p* < 0.05 with 95% confidence intervals.

**Figure 7 ijms-23-11560-f007:**
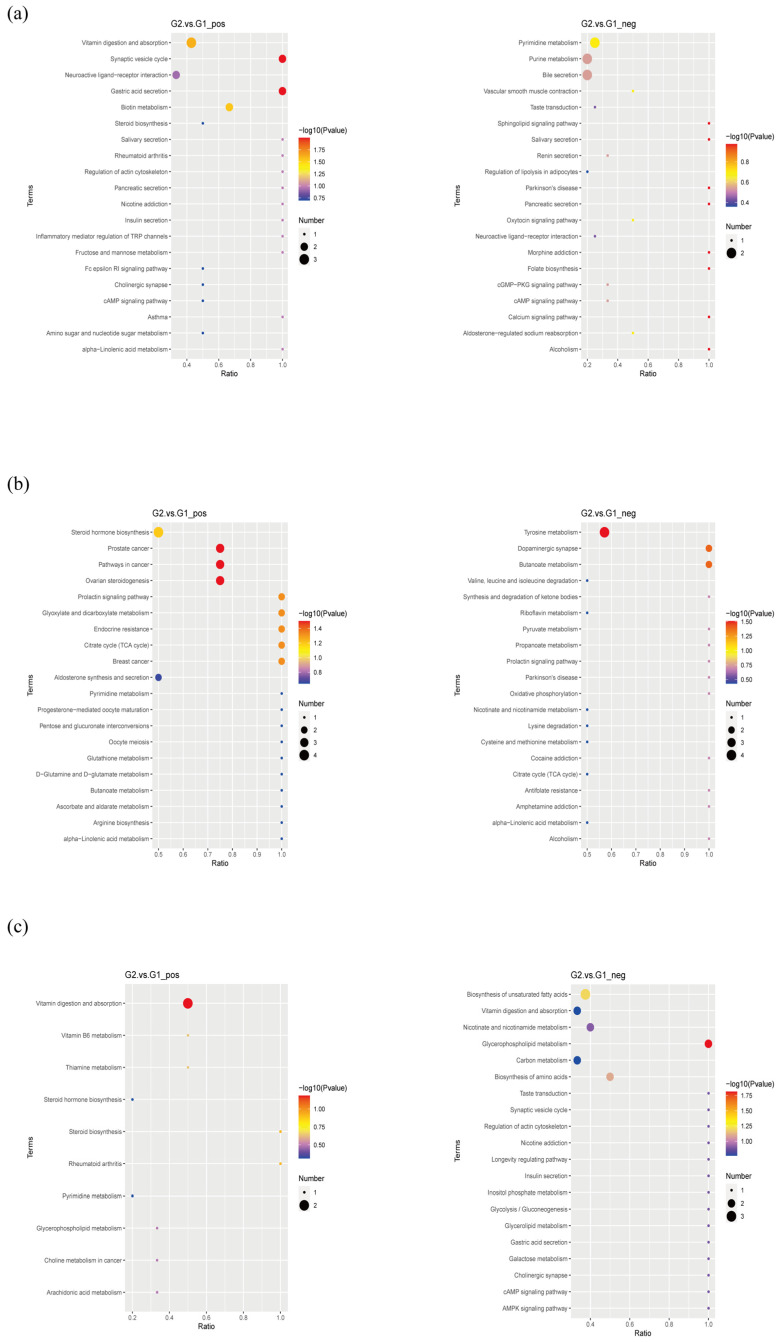
Scatter plots for the KEGG pathway enrichment of the changed metabolites between each compared group: (**a**) feces, (**b**) serum, (**c**) hippocampus.

**Figure 8 ijms-23-11560-f008:**
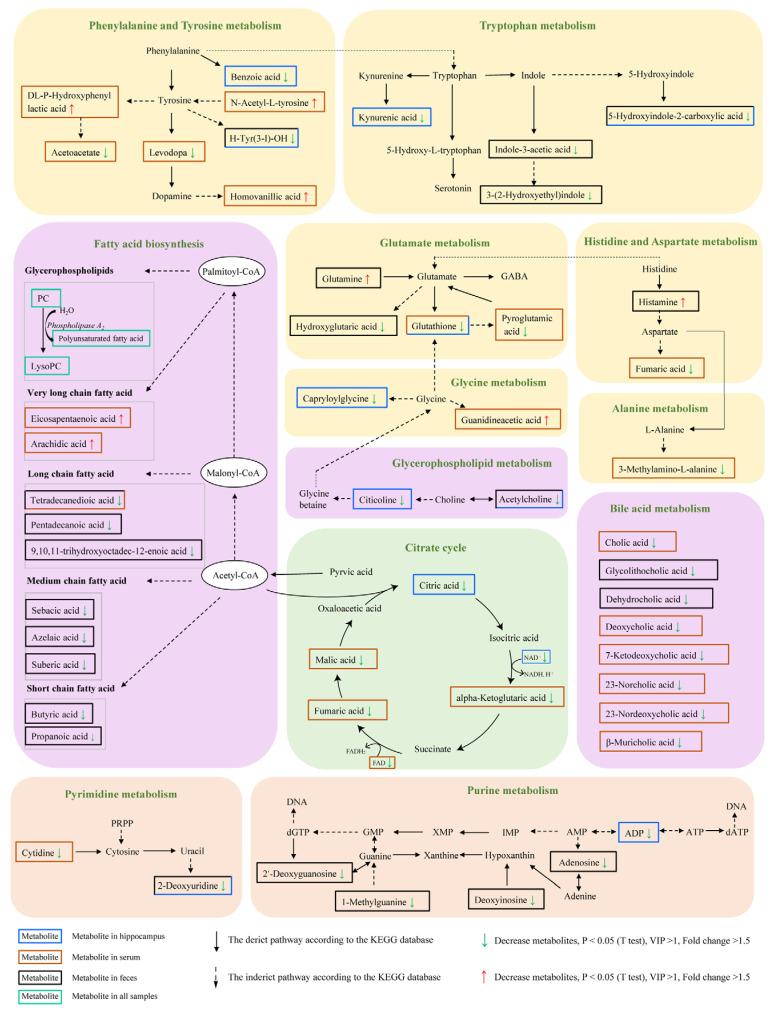
KEGG pathway of the differential metabolites between PAP and WT mice (Fold change >1.5, VIP >1, *p* < 0.05 *t* test).

**Figure 9 ijms-23-11560-f009:**
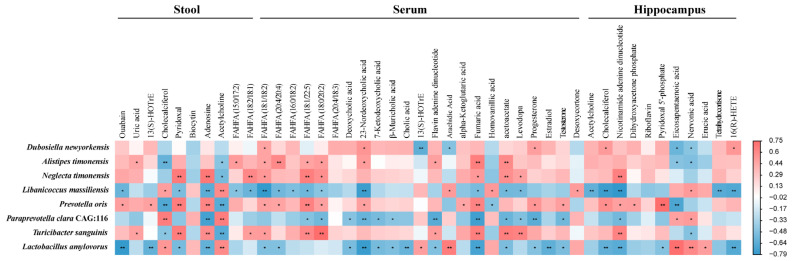
Heatmap showing Spearman’s correlation coefficients between the gut microbiota and metabolites in stool, serum, and hippocampus.

**Figure 10 ijms-23-11560-f010:**
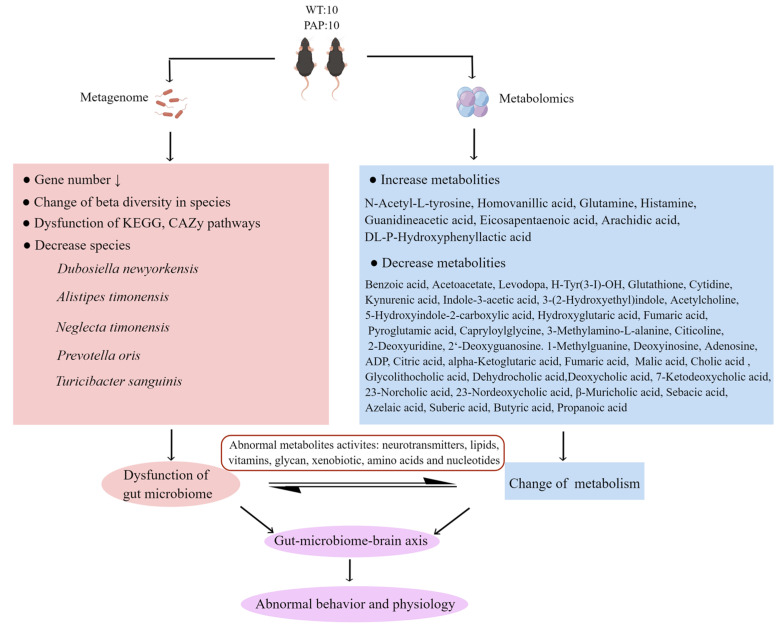
The summary of gut microbiota composition and metabolism analysis between AD and WT.

**Table 1 ijms-23-11560-t001:** The statistical table of metagenomics sequencing.

Sample Name	Raw Date	Clean Date	GC(%)	Effective (%)
WT1	6248.17	6238.41	47.17	99.844
WT2	6055.95	6048.28	47.76	99.873
WT3	6852.46	6840.39	45.97	99.824
WT4	6040.52	6031.72	45.10	99.854
WT5	6205.41	6196.22	44.73	99.852
WT6	6419.97	6411.56	48.30	99.869
WT7	6037.43	6027.86	49.06	99.841
WT8	6528.44	6521.04	49.18	99.887
WT9	6666.75	6649.82	48.47	99.746
WT10	6504.01	6489.21	47.29	99.772
PAP1	6182.81	6167.85	47.14	99.758
PAP2	5993.42	5982.55	48.96	99.819
PAP3	6212.57	6199.63	48.05	99.792
PAP4	6600.03	6574.39	47.45	99.612
PAP5	6887.99	6871.95	47.65	99.767
PAP6	6285.34	6270.28	48.46	99.760
PAP7	6665.45	6651.59	48.60	99.792
PAP8	6831.09	6814.69	45.81	99.760
PAP9	6142.11	6124.22	48.05	99.709

## Data Availability

The dataset analyzed in this research is available via reasonable email request to corresponding author.

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
