# Peer review of "Comparative Metagenomics and Metabolomes Reveals Abnormal Metabolism Activity Is Associated with Gut Microbiota in Alzheimer’s Disease Mice"

_ijms, 2022, doi:10.3390/ijms231911560_

Round 1
Reviewer 1 Report
The authors have studied the gut microbiome and metabolome between APPswe/PS1E9 (PAP) mice with cognitive decline and age-matched controls. The manuscript is interesting and well written, the experimental design is appropriate, and the methods and results are clearly described. However, minor concerns should be addressed before the manuscript can progress further. I hope the authors find these comments helpful in maximizing the impact of their work.
1- In the abstract, The authors should better describe the background and the conclusions of their study.
2-The acronym (AD) used in the introduction should be spelled out.
3- Most annotations and labels in the figures are not readable, so I suggest improving the quality of the figures to increase the readability.
4- In lines 139-140, this description is characteristic of discussion.
5- In line 570, please remove the dot before the reference.
6- The data Availability Statement is missing, and the authors should provide information about the availability of datasets used in the current study.
Author Response
Q1. In the abstract, the authors should better describe the background and the conclusions of their study.
R: Thank you for your kind suggestions. We have added and rephrased the background and conclusion in the abstract. And added the following sentences ‘In order to understand the mechanism of gut microbiota in AD, it is necessary to clarify the rela-tionship between gut microbiota and metabolites.’ in background and ‘Therefore, abnormal metabolism activity is associated with gut microbiota in Alzheimer’s disease mice.’ in conclusion, respectively.
Q2. The acronym (AD) used in the introduction should be spelled out.
R: Sincere gratitude to you for your meticulous advice. We have added the full name before the first “AD”.
Q3. Most annotations and labels in the figures are not readable, so I suggest improving the quality of the figures to increase the readability.
R: Thank you for your timely advice. According to your advice, we have re-inserted all figures properly to increase quality. Besides, we uploaded each original figure in order to better read these figures.
Q4. In lines 139-140, this description is characteristic of discussion.
R: Thank you for your kind reminder. This sentence is indeed inappropriate and we have deleted it.
Q5. In line 570, please remove the dot before the reference.
R: Sorry for the careless mistake. We have removed the dot before the reference.
Q6. The data Availability Statement is missing, and the authors should provide information about the availability of datasets used in the current study.
R: It’s very kind of you to have reminded us of this issue. Due to the datesets are too large more than 1000Gb, so if someone is interested in these date, please send email to corresponding author. Data Availability Statement is added at the end of the main text.
Reviewer 2 Report
Here are comments:
1. Browse through the manuscript as there are some typos, and be consistent with the words. For example:16S RNA or 16SRNA.
2. The experiment was done on females so why was prostate cancer looked into even results were shown significant change between the two groups.
3. The behavior test was done on female mice, in the introduction please include Alzheimer in prevalence in females. Females display a two-fold increase in the incidence of Alzheimer's compared to males. (doi: 10.31887/DCNS.2016.18.4/cepperson)
4. Include a diagram of your experiment design, for the ease of understanding of the readers.
5. Please discuss the histochemical data and its association with gut microbiota in PAP mice.
6. Also, did you also look into the association of behavior with metabolomics and metagenomics, and gut microbiome profile?
Author Response
Q1. Browse through the manuscript as there are some typos, and be consistent with the words. For example:16S RNA or 16SRNA.
R: Sorry for the careless mistake. I have re-checked the entire text to make sure all typos are corrected.
Q2. The experiment was done on females so why was prostate cancer looked into even results were shown significant change between the two groups.
R: Although women do not have prostate, not to mention getting prostate cancer, yet some prostate-cancer-related-metabolites, such as testosterone, also exist in women. And the metabolic pathway of prostate cancer will not be automatically removed by the software during analysis and plotting, which leads to this error. However, this error is meaningless.
Q3. The behavior test was done on female mice, in the introduction please include Alzheimer in prevalence in females. Females display a two-fold increase in the incidence of Alzheimer's compared to males. (doi: 10.31887/DCNS.2016.18.4/cepperson)
R: Thank you for this valuable and convenient advice. Adding this reference is very helpful.
Q4. Include a diagram of your experiment design, for the ease of understanding of the readers.
R: Thank you for your pointed advice. As you have suggested, we do have considered the necessity of displaying our experiment design. It was primarily demonstrated in Figure 10, despite the fact that some subsidiary experiments were not displayed to ensure the clarity of the design.
Q5. Please discuss the histochemical data and its association with gut microbiota in PAP mice.
R: Thank you so much for this pointed advice. The potential association between histochemical date and gut microbiota is indeed an intriguing and important aspect. And related discussions were added and highlighted in yellow.
Q6. Also, did you also look into the association of behavior with metabolomics and metagenomics, and gut microbiome profile?
R: This research is the preliminary result of our whole research, and we will try to clarify the association between them in our follow-up research.